# Mosquito Vectors (Diptera: *Culicidae*) and Mosquito-Borne Diseases in North Africa

**DOI:** 10.3390/insects13100962

**Published:** 2022-10-20

**Authors:** Amira Nebbak, Lionel Almeras, Philippe Parola, Idir Bitam

**Affiliations:** 1Centre de Recherche Scientifique et Technique en Analyses Physico-Chimiques (CRAPC), BP 384, Zone Industrielle, Bou-Ismail 42004, Algeria; 2Aix Marseille University, IRD, AP-HM, SSA, VITROME, 13005 Marseille, France; 3Unité Parasitologie et Entomologie, Département Microbiologie et Maladies Infectieuses, Institut de Recherche Biomédicale des Armées, 19-21 Boulevard Jean Moulin, 13005 Marseille, France; 4IHU-Méditerranée Infection, 13005 Marseille, France; 5École Supérieure en Sciences de l’Aliment et des Industries Agroalimentaire d’Alger, Oued Smar 16059, Algeria

**Keywords:** *Culicidae*, mosquitoes, vectors, mosquito-borne diseases, Algeria, Egypt, Libya, Morocco, Tunisia, North Africa

## Abstract

**Simple Summary:**

People around the world, as well as animals, are at risk of being affected by infectious diseases transmitted by mosquitoes. Indeed, in addition to being pests, some species of mosquitoes have the ability to carry pathogens from one host to another. In the North Africa region, approximately 80 species are currently recorded. In this review, we focus only on mosquitoes of medical and veterinary importance (26 species), providing a concise update of the literature on species and the diseases they carry (two parasitic and five viral infections).

**Abstract:**

Mosquitoes (Diptera: *Culicidae*) are of significant public health importance because of their ability to transmit major diseases to humans and animals, and are considered as the world’s most deadly arthropods. In recent decades, climate change and globalization have promoted mosquito-borne diseases’ (MBDs) geographic expansion to new areas, such as North African countries, where some of these MBDs were unusual or even unknown. In this review, we summarize the latest data on mosquito vector species distribution and MBDs affecting both human and animals in North Africa, in order to better understand the risks associated with the introduction of new invasive mosquito species such as *Aedes albopictus*. Currently, 26 mosquito species confirmed as pathogen vectors occur in North Africa, including *Aedes* (five species), *Culex* (eight species), *Culiseta* (one species) and *Anopheles* (12 species). These 26 species are involved in the circulation of seven MBDs in North Africa, including two parasitic infections (malaria and filariasis) and five viral infections (WNV, RVF, DENV, SINV and USUV). No bacterial diseases have been reported so far in this area. This review may guide research studies to fill the data gaps, as well as helping with developing effective vector surveillance and controlling strategies by concerned institutions in different involved countries, leading to cooperative and coordinate vector control measures.

## 1. Introduction

Mosquitoes are vectors of many infectious diseases caused by parasitic and viral agents such as lymphatic filariasis, chikungunya and Zika, which cause important pathologies in humans and animals [1]. According to some recent observations, they may also transmit bacteria such as *Rickettsia* (*R*.) *felis* [2]. The impact of these diseases on public health is significant. Considering malaria only, 229 million cases were reported worldwide in 2019, with 409,000 deaths. The African regional disease accounts for 90% of the cases and deaths occurring yearly [3]. However, all these cases are concentrated in sub-Saharan areas [4]. The North Africa countries, including Algeria, Egypt, Libya, Morocco, and Tunisia, suffer less from mosquito-borne diseases (MBDs). Nevertheless, according to modeling studies [5], climate change could promote the transmission risk of many MBDs, without mentioning the threat posed by indigenous vector-competent mosquito species to temperate regions [6].

Similarly to European countries [7], cases of the mosquito-borne West Nile fever have been reported in North Africa [8]. Furthermore, this region shares borders with Rift Valley fever-endemic countries such as Niger and Mauritania [9,10]. Therefore, for better implementation of vector control measures and MBD surveillance, a single health approach among North Africa members is needed. Currently, the MediLabSecure project aims to establish an integrated network covering the fields of human virology, animal virology, medical entomology and public health in nineteen non-European Union countries in the Mediterranean and Black Sea regions, including the five countries from the North Africa region [11]. This network has recently proposed guidelines for designing effective and sustainable entomological surveillance systems in order to improve preparedness and response capacities to mosquito-borne viral diseases [12]. Likewise, VectorNet project supports the collection of data on arthropod vectors and related pathogens across Europe and the Mediterranean basin, including mosquitoes [13].

After the pioneering entomological works realized by the Sergent brothers [14] and Senevet and Andarelli [15,16], forty-years later, Brunhes et al. developed useful software for identifying *Culicidae* in Mediterranean Africa [17]. Recently, Robert et al. provided an updated distribution chart of the Euro-Mediterranean mosquito species including Algeria, Egypt, Lybia, Morocco and Tunisia [18]. Subsequently, in Algeria for example, inventories of mosquito populations were carried out in different dispersed regions, as well as studies on mosquito susceptibility to bimolecular substances extracted from plants or Fungi, focusing mainly on *Cx. pipiens* sensu lato (s.l.) [19,20]. However, very recently, Merabti et al. provided an interesting, updated checklist of the mosquito species present in Algeria [21]. In North Africa, mosquito epidemiological studies remain scarce, with few recent works investigating pathogen detection, such as *Dirofilaria immitis* parasite [22] and the West Nile virus [23,24]. Emerging vector-borne diseases have been reported in temperate regions [25,26]. In this review, we aim to update the status of MBDs and their vectors in North Africa. We collated MBDs which occurred this last decade in North Africa and highlighted the most recent advances in entomological studies. The risk of the introduction of new mosquito vectors with their associated pathogens promoting the emergence of new MBDs is discussed. Vector control strategies are also evoked. Finally, state-of-the-art mosquito identification techniques are presented.

## 2. Methodology

For this review, data were compiled from published articles retrieved from NCBI’s PubMed and Google Scholar, using the keywords: mosquito, *Culicidae*, name of countries from North Africa (e.g., Algeria), mosquito species name (e.g., *Ae*. *albopictus*) and disease/pathogen (West Nile fever/*Plasmodium* sp.). We also used Research Gate, a social networking site for scientists and researchers, reviewed by retrieving data made available by the authors themselves, such as articles and personal communication, in addition to grey literature from international and national institutions (World Health Organization WHO, Pasteur Institute). More than 300 publications were recovered using the keywords above, focusing essentially on the last two decades; around 100 were then excluded, either because the article was not fully accessible, or to avoid redundancies.

## 3. Salient Data on North Africa

The five North Africa countries, Algeria, Egypt, Libya, Morocco, and Tunisia, included in this review, cover an area of 5,605,641 km^2^, and have a total population of 184,686,758 inhabitants. More than half of this population originates from Egypt. The climate is varied, and divided between a Mediterranean climate in the northern part, an oceanic climate in part of Morocco, and a Sahara desert climate, according to the Köppen-Geiger classification [27]. The region contains several wetlands, of which 201 are protected and recognized by the Ramsar Convention [28], such as Tonga Lake in Algeria and Afennourir Lake in Morocco, providing favorable conditions for mosquito development. The region is crossed by major migratory bird flyways [29], which may potentially be involved in the introduction and spread of viruses [30]. Small ruminants such as sheep and goats dominate the livestock population, due to cultural traditions, but cattle and camels are also widely bred. Animal farming and livestock production are characterized by a significantly extensive nomadic system, which can increase the risk of VBDs spreading in North Africa [31,32]. A population of bats exists in North Africa, and these mammals are known as virus reservoirs [33,34]. All these ecological riches make the North Africa region a meeting place for vectors, pathogenic agents and reservoirs.

## 4. Morphology, Life Cycle, Taxonomy, Genomics and Identification Methods of *Culicidae*

Mosquitoes (Diptera: *Culicidae*) are winged, thin-bodied insects, with long thin legs. They are characterized by a long scaled proboscis [35]. Only the females are hematophagous; the blood meal taken on vertebrate mammalians is indispensable for the egg maturation in their ovaries. In the *Culicidae* family, the head, thorax and abdomen are covered with scales and setae; the extent and distribution of coverage is genus-specific. The legs, wing margins, and wing veins are typically covered with scales [35]. The life cycle of all mosquito species is completed in four distinct stages: egg, larva and pupa, occurring in water, and finally the imago adult flying stage. The female mosquito lays her eggs directly on water or on moist substrates that may be flooded with water. The egg then hatches into the larva, which goes through four developmental stages. Next, the larva transforms into the pupa, a resting, non-feeding stage, during which metamorphosis occurs until the emergence of the adult flying mosquito [36]. The classification of *Culicidae* is evolving, according to technological advances in molecular and genetic biology. The family *Culicidae* is divided into two sub-families: *Culicinae* and *Anophelinae*. Currently, 3601 mosquito species and sub-species have been reported worldwide, and changes have been made in the Aedini tribe taxonomy [37]. The abbreviations of genera (two letters) and sub-genera (three letters) are regulated under a standardized nomenclature system [38].

Currently, entire genomes from solely 36 mosquito species are available in GenBank, and among them, six mosquito species have been reported in the North Africa region (Table 1). The identification of specimens at the species level is particularly essential to knowing exactly which mosquito species are involved in the transmission cycle of a vector disease. Most of the studies conducted in North Africa on mosquito species used morphological keys; this remains the standard method to identify collected species. This technique does not require expensive resources, but great expertise is necessary for accurate identification. The integrity and correct storing of the specimen preventing the loss of key morphological criteria are decisive factors to succeed in identification. Molecular techniques such as PCR, sequencing and barcoding, have also been successfully applied, but the relatively long processing time and the cost of reagents could be limiting factors for their large application in most North African laboratories [39,40].

Recently, a rapid and simple method of amplifying DNA with high specificity and efficiency, named loop-mediated isothermal amplification (LAMP), demonstrated its performance for mosquito identification in the field, using a high-throughput format [46]. The matrix-assisted laser desorption/ionization time-of-flight mass spectrometry (MALDI-TOF MS) profiling method, a proteomics approach, was successfully applied to the identification of adult mosquito specimens, larvae, and eggs, as well as exuviae [47,48,49,50,51,52]. Very recently, an Algerian study introduced this tool for the identification of field-caught species [53]. Other classification methods such as CNN (convolutional neural network), based on extracting features from mosquito images, have also produced promising results [39]. Recent observations recommended combining all these methods from the perspective of an integrated taxonomy [40].

## 5. *Culicidae* Fauna of Medical and Veterinary Importance in North Africa

Approximately eighty-three mosquito species were reported in North Africa by Robert et al. in 2019 [18], including 59 *Culicinae* and 24 *Anophelinae* (Appendix A). Among these, 26 species were confirmed as potential vectors for transmitting pathogens to humans and/or animals (Table 2). Details of these mosquito species are now inventoried.

### 5.1. Sub-Family Culicinae

In this section, species of three genera, *Aedes*, *Culex* and *Culiseta,* are presented.

#### 5.1.1. *Aedes* (Meigen)

Twenty-three *Aedes* species have been reported in North Africa (Appendix A). Five of them are described as vectors of viral and parasitic diseases, including *Ae. (Stegomyia) albopictus* (Skuse, 1895), *Ae. (Stegomyia) aegypti* (Linnaeus, 1762), *Ae. (Ochlerotatus) caspius* (Pallas, 1771), *Ae. (Ochlerotatus) detritus* (Haliday, 1833), and *Ae. (Aedimorphus) vexans* (Meigen, 1830) (Table 2). The *Aedes* genus encompasses some of the most feared vectors responsible for arbovirus transmission, such as Yellow Fever (YF), dengue, chikungunya, Zika, Rift Valley Fever (RVF) and West Nile (WN), as well as some parasites (e.g., dog heartworm, filariae) and bacteria [35]. Mosquitoes of this genus have worldwide distribution; species are found in temperate zones, tropical zones and forest zones, in rural and urban areas, where they can be vicious biters and serious pests to both people and livestock. The selection of an aquatic habitat is dependent of the *Aedes* species. They range from large areas, such as marshes and ground pools, including snow-melt pools in arctic and subarctic areas, to very small collections of water such as tree-holes, bamboo stumps, leaf axils and rock-pools, or man-made ones such as water-storage pots, tin cans and tires [65,71].

***Aedes (Stegomyia) albopictus* (Skuse, 1895)**, is an aggressive daytime biting mosquito. It present an exceptional expansion capability. Native to the East Asia region, it previously colonized islands of the western Pacific and the Indian Ocean, to expand to every continent of the world, with the exception of Antarctica [72]. This invasive Asian mosquito is commonly called the tiger mosquito, due to its body being zebra-striped. The first description of *Aedes (Stegomyia) albopictus* (Skuse, 1895) (=*Stegomyia albopicta* in the new classification), in North Africa, occurred in Tizi Ouzou province, Algeria, during an entomological program targeting sandflies in August 2010 [54]. It was reported for the second time in the same region four years later [73]. In 2015, its establishment in the Mediterranean Africa area was confirmed in Oran province in west Algeria, where for the first time, in addition to imago stages, immature stages were also described [55]. The same year, in an urban area of Rabat district, Morocco, an Algeria neighboring country, larval of *Ae. albopictus* were identified, for the first time [74]. The involvement of *Ae. albopictus*, newly introduced to Morocco, in local transmission of dengue, chikungunya, Zika and YF to humans, is a likely scenario [75]. More recently, *Ae. albopictus* has been reported in Algiers [76,77], Annaba [78] and Tunisia [79]. Its establishment in Egypt or Libya has not been yet reported. However, a recent study predicted the future distribution of *Ae. albopictus*, and identified Egypt and Libya as regions at risk of invasion [80].

In addition to being a competent vector for several viruses [81], it is suspected of enhancing the inter-animal and zoonotic transmission of filarial nematodes [82]. Its role in the natural transmission of *Dirofilaria repens* has been observed in Italy [83]. Additionally, *Ae. albopictus* was found to be naturally infected by *R. felis* in Gabon [84]. Its great adaptability and capacity to transmit multiple pathogens make this mosquito one of the main vectors of infectious diseases. The setting up of surveillance and control programs is required, to prevent outbreaks.

***Aedes (Stegomyia) aegypti* (Linnaeus, 1762)**, the Egyptian tiger mosquito, is a major vector of YF, but also dengue and other viral agents [65]. It can be confused with *Ae. albopictus*, due to their similar morphology, but a detailed check of the picture on the thorax is helpful for their distinction. *Ae. aegypti* is native to West Africa, and was originally a zoophilic mosquito living in African forest environments. Subsequently, several factors have contributed to its adaptation to the urban environment and worldwide invasion, in particular genetic predisposition, drought-resistant eggs, preferences for human blood, and larval development in man-made container habitats [85]. This species has been reported in the five countries of North Africa [17]. Despite its complete eradication in 1967 in Egypt [56], this mosquito re-emerged forty years later, in southern Egypt, in the Aswan [86] and Toshka localities [56]. This re-emergence raises questions about an epidemiological risk, as this mosquito is a vector of arboviruses, such as YF, dengue, CHKV and Zika.

***Aedes (Ochlerotatus) caspius* (Pallas, 1771)** is widely distributed in the North Africa area. *Ae. caspius* is considered as a potential vector of RVF [87], and is involved in the circulation of the WN and Usutu viruses (USUV) in Italy [88]. In Egypt, *Ae. caspius* specimens infected by *Dirofilaria repens* in the El Nikhila locality have been reported [61]. Nevertheless, experimental infections conducted on Tunisian *Ae. caspius* specimens revealed that this species was unable to transmit an Asian genotype of ZIKV [89].

***Aedes (Ochlerotatus) detritus* (Haliday, 1833)**. Reported in the five studied countries [18], *Ae. detritus* is a potential vector of RVF [87]. *Ae. detritus* was found to be naturally infected by USUV in Italy [88] and experimentally competent for the transmission of Japanese Encephalitis Virus (JEV) and WN Virus (WNV) in the UK [90,91]. A study conducted in Tunisia showed that *Ae. detritus* was unable to transmit an Asian genotype of ZIKV [89].

***Aedes (Aedimorphus) vexans* (Meigen, 1830)**, a flood water mosquito, has been detected in Algeria, Libya, Morocco and Tunisia [18]. It is a potential vector of RVF [87,92], and could serve as a potential vector of ZIKV [93]. In Serbia, it was found to be naturally infected by *D. immitis* and *D. repens* [94]. Recently, Anderson et al. reported that a Connecticut strain of *Ae. vexans* was competent to transmit WNV both horizontally to suckling mice and vertically to its progeny, in experimental conditions [95].

#### 5.1.2. *Culex* Linnaeus

Twenty-five *Culex* species have been described in North Africa (Appendix A). Among them, eight species were described as vectors or potential vectors of pathogens. They include *Culex (Culex) pipiens pipiens* (Linnaeus 1758), *Culex (Culex) pipiens quinquefasciatus* (Say 1823), *Culex (Culex) antennatus* (Becker 1903)*, Culex (Culex) perexiguus* (Theobald 1903), *Culex (Culex) univittatus* (Theobald 1901), *Culex (Culex) theileri* (Theobald 1903), *Culex (Barraudius) modestus* (Ficalbi 1889) and *Culex (Barraudius) pusillus* (Macquart 1850) (Table 2).

Species of the *Culex* genus are found in both temperate and tropical regions; the larval habitats are very varied, including permanent or temporary pools, flooded land, irrigated crops, riverbanks, hollows of trees or axils of sheathing leaves, as well as man-made container-habitats such as tin cans, water receptacles, bottles and water-storage tanks. Adults mainly bite at night-time; trophic preferences are also varied, but remain mainly oriented towards birds or mammals. Some species such as *Cx. pipiens* and *Cx. quinquefasciatus* are associated with urbanization, especially towns with poor and inadequate drainage and sanitation, and they have the ability to lay in highly polluted waters [65,71].

***Culex (Culex) pipiens pipiens* (Linnaeus, 1758)** is the most widely distributed mosquito species in urban and sub-urban temperate regions, including North Africa. It is a member of the *Cx. pipiens* complex, which includes four other species: *Cx. quinquefasciatus* Say, *Cx. pallens* Coquillett, *Cx. globocoxitus* and *Cx. autralicus* [96]. *Cx. pipiens* has two molecular forms that can be distinguished using multiplex PCR, based on the microsatellite locus CQ11 [97,98,99]. The hybrid form was described for the first time in North Africa in Morocco, where *Cx. pipiens* form *pipiens* and form *molestus* were found in urban, suburban and rural habitats [97]. In Algeria as well, the genetic form *Cx. pipiens molestus* lives along with the *pipiens* form in addition to hybrids (*molestus*/*pipiens*) collected from different environments [98,99]. In Tunisia, a recent study aimed to identify *Cx. pipiens* forms occurring in the country, as well as autogenic behavior. Among the specimens classified as *Cx. pipiens* sl, 33.5%, 30.8% and 35.6% were identified as forms *pipiens*, *molestus*, and hybrids, respectively, using the microsatellite CQ11. In this study, the authors confirmed that *Cx. pipiens* forms can share the same site, regardless of breeding site or habitat, without competitive exclusion, and highlighted the abundance of autogeny [57]. Interestingly, Shaikevich et al. revealed the first evidence of the presence of hybrids between *Cx. quinquefasciatus* and *Cx. pipiens*, as well as cytoplasmic introgression of *Cx. quinquefasciatus* into *Cx. pipiens* as a result of hybridization events in the Mediterranean region, including Morocco (Tangier). These cases of hybridization can change the properties of vectors, due to the genetic contribution of the more anthropophilic *Cx. quinquefasciatus* [62].

It has been proved experimentally that the *Cx. pipiens* populations collected in Algeria, Morocco and Tunisia are efficient vectors of WN, and to a lesser extent RVF viruses, isolated from a horse in Camargue and a human case in Bangui (Central African Republic), respectively [87,100]. *Cx. pipiens* is also the vector of *Dirofilaria immitis* filariasis. In the Sohag governorate, Egypt, *Cx. p. molestus* was found to be infected by *Wuchereria bancrofti* and *D. immitis*, based on histological examination and PCR tests [101], and three filarial nematodes; *Wu. bancrofti*, *D. immitis* and *D. repens*, in an endemic region (Assiut Governorate), based on single PCR and multiplex PCR [61]. These studies concluded that molecular tools are able to differentiate *Dirofilaria* species infecting different mosquito species.

***Culex (Culex) pipiens quinquefasciatus* (Say, 1823)** is the most prevalent *Culex* species in tropical and sub-tropical areas. It was reported uniquely in Morocco and in Tinzouatine, a village in southern part of Algeria in 2008 [59,62]. However, modeling predicted its geographic distribution change due to global warming, showing all the coastal area of the North Africa region becoming a suitable *Cx. quinquefasciatus* establishment [102]. *Cx. quinquefasciatus* is the vector of WNV, Saint Louis encephalitis (SLE), RVF Virus (RVFV) and lymphatic filariasis [102]. Experimental studies of vector competence have confirmed that *Cx. quinquefasciatus* can disseminate and transmit ZIKV [103].

***Culex (Culex) antennatus* (Becker, 1903)** has been described in Algeria, Egypt, Morocco and Tunisia [18]. This mosquito is a vector of WNV and RVFV, as well as *Dirofilaria repens* [104,105]. In Egypt, *Cx. Antennatus* was found to be infected by *D. repens* in an endemic region [61].

***Culex (Culex) perexiguus* (Theobald, 1903)**. Distributed in the five studied countries [18], *Cx. perexiguus* is a vector of WNV and potential vector of RVFV [87]. Recently, a pool of collected *Cx. perexiguus* in the oasis of Aougrout, a province of Timimoun in the Algerian Sahara, was found to be positive for the lineage 1 of WNV [24,76]. A newly discovered flavivirus named Barkedji virus was identified in *Cx. perexiguus* mosquitoes in Israel [106].

***Culex (Culex) univittatus* (Theobald, 1901)*,*** have been reported in Egypt, Morocco and Tunisia, but a definitive confirmation is pending [18]. *Cx. univittatus* is considered as vector of WNV and Sindbis virus [107,108].

***Culex (Culex) theileri* (Theobald, 1903)**, a potential vector of WNV and RVFV [87], have colonized the five North Africa countries [18]. Several new flaviviruses have been isolated and identified using metagenomics in *Cx. theileri* [109].

***Culex (Barraudius) modestus* (Ficalbi, 1889)**, reported in Algeria and Morocco, is a vector of the WNV, USUV, Tahyna, and Lednice virus [88].

***Culex (Barraudius) pusillus* (Macquart, 1850)** is a vector of *Dirofilaria immitis*. In dirofilarioses endemic areas of Egypt, *Cx. pusillus* was found infected by *D. immitis* [61].

#### 5.1.3. *Culiseta* (Felt)

Six species of *Culiseta* have been reported in North Africa (Appendix A), among which one is known to be an arbovirus vector. Reported in Algeria, Morocco and Tunisia, ***Culiseta (Culiseta) annulata* (Schrank, 1776)** was suspected to be a vector of the Tahyna virus. The species breeds in a wide variety of permanent and semi-permanent habitats, including natural and artificial water, in both open and shaded situations. Larvae can be found in stagnant pools, ponds, ditches, water troughs, and other artificial containers, such as barrels collecting rain water [35].

### 5.2. Sub-Family Anophelinae

#### *Anopheles* (Meigen)

Among the 24 anopheline species indexed in North Africa countries, half of them (*n* = 12) were considered as vectors of pathogenic agents (Table 2), including *Anopheles (Anopheles) labranchiae* (Falleroni, 1926),* Anopheles (Anopheles) algeriensis* (Theobald, 1903), *Anopheles (Cellia) multicolor* (Cambouliu, 1902), *Anopheles (Cellia) sergentii* (Theobald, 1907),* Anopheles (Cellia) superpictus* (Grassi 1899)*, Anopheles (Cellia) coluzzii* (Coetzee & Wilkerson, 2013),* Anopheles (Cellia) pharoensis* (Theobald, 1901), *Anopheles (Cellia) stephensi* (Liston, 1901), *Anopheles (Anopheles) coustani* (Laveran, 1900), *Anopheles (Anopheles) claviger* s.s. (Meigen, 1804), *Anopheles (Anopheles) sacharovi* (Favre, 1903) and *Anopheles (Cellia) arabiensis* (Patton, 1905).

Species of the *Anopheles* genus are the only ones that carry the *Plasmodium* parasite species. They have an almost worldwide distribution [110]. Breeding sites are very varied, according to the species; *Anopheles* can develop in fresh or brackish water, sunny or shaded sites, stagnant or running water [110]. *Anopheles* usually takes blood meals nocturnally or at dusk, from humans, mammals, and, rarely, birds [65].

***Anopheles (Anopheles) labranchiae* (Falleroni, 1926).** It is the unique member of the *An. maculipennis* group present in Algeria, Morocco and Tunisia [64]. It is a malaria vector in Algeria, Morocco and Tunisia (especially in northern governorates) [64,111]. It has some synonyms, such as *An. sicaulti* in Morocco [112]. During an entomological study conducted in Tunisia, *An. labranchiae* was collected in the northern and central parts of the country [64].

***Anopheles (Anopheles) algeriensis* (Theobald, 1903).***An. algeriensis* is considered as a secondary vector of malaria due to its exophilic behavior and scarcity [35]. *An. algeriensis* is rarely encountered in villages, and even in the open field an abundant population is rarely found [35]. It was collected at a low density from only 8% of larval habitats during a study conducted in Tunisia in 2012, suggesting that this species is scarce and does not presently and did not historically pose a risk of malaria transmission [64].

***Anopheles (Cellia) multicolor* (Cambouliu, 1902).** This *Anopheles* species is a potential vector of RVF [87] and *Plasmodium* in Algeria and Tunisia [64,69]. In some areas, *An. multicolor* is the dominant species. In 2012, an entomological study revealed its presence in 83.3% of the positive breeding places in the southern part of Tunisia [64]. Recently, during an entomological survey, none of the *An. multicolor* females was found to be infected by *Plasmodium*, using DNA screening in Al-Adwa village, the Aswan governorate of Egypt [113].

***Anopheles******(Cellia) sergentii* (Theobald, 1907)**, is a vector of *Plasmodium* in Algeria, Egypt, Morocco and Tunisia [64,69,113,114]. A study conducted in 2012 in Tunisia revealed its presence in 83.3% of the positive breeding places visited in the south of the country [64]. Recently, during an entomological survey conducted in Al-Adwa village, Aswan governorate in Egypt, *An. sergentii* females were found negative for *Plasmodium* DNA screening [113].

***Anopheles (Cellia) coluzzii*****(Coetzee and Wilkerson, 2013).***Anopheles gambiae* s.l. complex comprises at least nine species; two of them were reported in North Africa, namely *An. colluzzi* and *An. arabiensis*. These species are considered as the most important malaria vectors in sub-Saharan Africa. In 2007, *An. coluzzii* (cited as the Mopti form of *An. gambiae*) was occasionally detected in southern Algeria during a malaria outbreak. They probably originated from Mali, a border region where *An. gambiae* is established [69]. Robert et al. indicated its presence as uncertain in Algeria [18].

***Anopheles (Cellia) arabiensis* (Patton, 1905)** is an important vector of malaria wherever it occurs [35]. Present in Egypt [18], this species was already likely to enter Algeria via Niger in the 1980s [115], but no study has reported its presence nowadays.

***Anopheles (Cellia) pharoensis* (Theobald, 1901).** It is a vector of malaria and different filarial nematodes in Egypt [61,116]. In Egypt, 0.08% of collected *An. pharoensis* were found to be infected by *Wu. bancrofti* and *D. repens* in the El Nikhila and El Matiaa localities, respectively [61].

***Anopheles (Cellia) stephensi* (Liston, 1901).** Reported only in Egypt, *An. stephensi* is an important vector of malaria in the Red Sea Coast [117]. It is also a vector of malaria on the Arabian peninsula and large cities in the Indian subcontinent [35].

***Anopheles (Anopheles) coustani* (Laveran, 1900).** The presence of *An. coustani* in Egypt, Morocco and Tunisia has been reported [18], whereas its detection in Algeria is controversial [21]. In addition to being a malaria vector, *An. coustani* was found infected by RVFV in Madagascar [105] and by ZIKV in Senegal [118].

***Anopheles (Anopheles) claviger s.s.* (Meigen, 1804)**, is considered as a secondary vector of malaria in North Africa [65]. This species had already been reported in Algeria, Libya, Morocco and Tunisia [18,66,67].

***Anopheles (Cellia) superpictus* (Grassi, 1899)**. Reported in Algeria, Egypt, Libya and Tunisia, this species is an important malaria vector in central Asia [35]. Nevertheless, it has not been involved yet in the transmission of malaria to humans in North Africa.

***Anopheles (Anopheles) sacharovi* (Favre, 1903)** is a vector of malaria, reported in the north-east of Algeria by Boudemagh et al. in 2013, based on morphological data [68]. Despite their presence in North Africa, some species are not involved in transmission cycles. However, their distribution and infectious status must be closely monitored, and special attention should be paid to *Ae. albopictus*.

## 6. Mosquito-Borne Diseases in North Africa

### 6.1. Parasitic Infections

#### 6.1.1. Plasmodium

The hematozoa responsible for malaria was discovered at Constantine, Algeria, in 1880, by Dr Alphonse Laveran. Four years later, he published the vector role of the mosquito in the transmission of the disease [119]. In 1902, the first effort for malaria control in the world was put in place in Algeria, thanks to the Sergent brothers. At the beginning of the 1960s, there was a marked increase in malaria cases in this country. The launching of an Algerian malaria eradication campaign in 1968, reduced significantly the number of malaria human cases in the following years, from 95,424 cases in 1960 to 30 cases in 1978. At that time, the northern part of the country was declared free of *Plasmodium (P.) falciparum*; only a few cases due to *P. vivax* persisted, in residual foci in the middle part of the country [120]. Malaria was eliminated from Libya in 1957 [121], from Tunisia in 1980 [64] and from Morocco in 1974 (only *P. falciparum*) [122]. However, at the beginning of the 1980s, an increase in imported malaria cases was observed [120,121,122]. North Africa countries continue to confront the problem of imported malaria, and the risk of reintroduction cannot be formally ruled out because of the possible infestation of local *Anopheles* with plasmodial strains from endemic countries [123]. Indeed, in November 2007, an outbreak of autochthonous malaria due to *P. falciparum* occurred in Tinzaouatine, southern Algeria. Genetic resistance to chloroquine was detected in the parasite [69]. In Morocco, a retrospective study conducted from 1997 to 2007, reported imported cases from 13 African countries [111,124]. In Egypt, an outbreak of *P. vivax* occurred in 2014 in the Aswan Governorate [125].

Moreover, the importation of human malaria cases in southern Europe has already been reported, notably in migrants, whatever their origin, having transited North-Africa countries [126]. The absence of new indigenous contamination since 2013 in Algeria led the WHO to issue malaria-free certification to this country in 2019 [127]. Nevertheless, despite the elimination of malaria, vigilance to prevent malaria outbreaks from countries of North Africa should be maintained by the development of monitoring and vector control programs [128].

#### 6.1.2. Filariasis

##### *Wuchereria Bancrofti*  

The pathogenic agent of the lymphatic filariasis or elephantiasis is listed among the neglected tropical diseases by the WHO (https://www.who.int/health-topics/neglected-tropical-diseases#tab=tab_1 (accessed on 1 September 2022)). Lymphatic filariasis was endemic in Egypt, but was treated with mass drug administration until reaching a prevalence of less than 1% [129]. To support the elimination programs, epidemiological studies were conducted in endemic areas, such as Giza and Qualioubiya, researching DNA from *Wu. bancrofti* in collected mosquitoes and in blood samples [130,131]. Through these field surveys, the predominant mosquito species were identified, and genetic diversity of *Wu. bancrofti* was found in blood samples [130]. Follow-up genetic and entomological studies with larger sample sizes are recommended. These findings may influence the elimination program outcome in endemic areas, by distinguishing related strains, and identifying the various ecological- and drug-related effects, as well as detecting the infectiousness and exposure of the vectors [130,131].

##### *Dirofilaria immitis* and *Dirofilaria repens*

Heartworm disease and subcutaneous dirofilariasis are two pathologies caused by the parasitic nematodes *D. immitis* and *D. repens,* respectively. The reservoir is the dog, while man is accidentally infected by mosquito bites. In Algeria, a pioneering serological study conducted in 2007 in Algiers reported a high prevalence of *D. immitis*, ranging from 18.5 to 24.5% in dogs [132]. Five years ago, a study conducted in the provinces of Algiers and Tizi Ouzou on living dogs revealed the presence of *D. immitis* in 1.4% of the dogs molecularly tested from Algiers, whereas no canine cases of *D. repens* were reported [22].

In Morocco, serological analyses revealed that 16.1% of dogs sampled were positive against *D. immitis* antigens in seven different locations of the country; higher rates were recorded in the northern central region [133]. Interestingly, in this study, *D. immitis* was found in association with *Anaplasma* spp. and *Ehrlichia* spp. in dogs [133]. In Tunisia, a prevalence of 14.5% and 3% for *D. immitis* and *D. repens*, respectively, were reported in dogs [134]. Except for studies conducted in Egypt, where these parasites were researched in mosquitoes, including *Cx. p. molestus*, *Cx. antennatus*, *Cx. pusillus*, *Ae. caspius*, *An. pharoensis* and *Cs*. *longiareolata* [61,101,135], all investigations of *Dirofilaria* infections targeted dogs. Testing mosquitoes found around positive dog cases is important for identifying the vector in the region of interest and for knowing the prevalence of infection, and consequently the risk of parasite transmission, to humans. Even if they are infrequent, human cases were registered in North Africa. Indeed, in Tunisia, several human cases of filarial infections due to *D. repens* have been reported, affecting different body parts, such as the skin [136,137], the upper lip [138], conjunctival infection from eyelids [139,140], peri-tendon from hand [141] or breast from two women [142]. In Egypt, antigens from *D. immitis* were detected in 3.4% of the cats sampled [143]. In this country, human cases associated with *D. repens* have also been reported as subcutaneous and pulmonary cases in the Assiut governorate [144].

### 6.2. Arboviruses

Arthropod-borne viruses (arboviruses) are viruses transmitted by hematophagous arthropods such as ticks, sandflies, biting midges and mosquitoes [145]. Mosquito-borne viruses (MBVs) have been estimated to cause over 100 million cases of human disease annually [6]. Several MBVs belonging to the genera *Alphavirus*, *Flavivirus* and *Bunyavirus* have been reported to occur in North Africa, infecting humans and other vertebrates. However, some arboviruses such as ZIKV, have never been reported in North Africa, even though their mosquito vectors are present. The non-detection of these arboviruses (ZIKV, CHIKV, etc.), could be attributed to the real absence of these pathogens or to insufficient investigations conducted in North Africa. Nevertheless, the local presence of mosquito vectors suggests that the risk of introduction and spreading of these diseases is realistic.

#### 6.2.1. West Nile Virus (WNV)

WNV (*Flaviviridae*, *Flavivirus*) is the most studied arbovirus in North Africa, due to its widespread presence and endemicity in this region. It was isolated for the first time in North Africa in Algeria in the region of Djanet in 1968 [146]. The primary isolation of WNV in Morocco occurred thirty years later [147]. Since 1973, in several southern and central regions of Algeria, serological surveys have demonstrated human exposure to WNV [146]. In 1994, an epidemic occurred in six oases in the Tinekouk community in the Timimoun region, in the central Sahara of Algeria [148]. Approximately 50 suspect human cases of WNV were reported, and eight deaths occurred. Epidemics among horses occurred in Morocco in 1996, 2003 and 2010 [8]. In 2011, a seroprevalence survey in humans showed that, among the 165 samples collected, 16 (approximately 10%) possessed specific antibodies against WNV [149].

Since then, the first human death due to WNV has occurred, in 2012, of a patient who sojourned in Algeria [150]. A study conducted in 2014 on the seroprevalence of WNV in an equid population (horses and donkeys) revealed the presence of the WNV in the northeastern wetland of Algeria, especially in the El Kala region [23]. The virus lineage circulating in Algeria was unknown [149] until recently, and field studies on mosquitoes found them positive for the lineage 1 of WNV [24,76]. A serological study conducted among equids in the Kébili governorate, southwest Tunisia, revealed that 42.3% were positive. This virus apparently co-circulated with USUV [151]. The virus was detected by RT-qPCR and isolated on Vero cells for the first time from *Cx. pipiens* mosquitoes in Tunisia.

Phylogenetic analysis revealed that the virus was from lineage 1, closely related to the Italian mosquito WNV strain [152]. In 2013, a serological study conducted in Libya on the Tripoli city population revealed that 2.75% of human blood samples tested had antibodies against WNV antigens [153]. Hachid et al. conducted a retrospective study targeting human sera collected in Algiers and the surrounding provinces. This work underlined the fact that 9.8% of the 164 collected sera had IgG against WNV, which was confirmed for the 6.7% of them harboring specific neutralizing antibodies [154].

A study conducted on yellow-legged gull eggs on the Mediterranean coast revealed that 3% of eggs were positive for *Flavivirus* antibodies in Jijel (Algeria), using the Elisa test. This low prevalence led the authors to suggest that it might reflect past *flavivirus* circulation at this site [30]. Evidence for WNV disease in other animals in North Africa has been confirmed [8].

In 2015, Benjelloun et al. published a detailed review on the epidemiology of WNV disease in northwest Africa, focusing on Algeria, Morocco and Tunisia [8]. Subsequently, a recent review summarized the current epidemiological situation of West Nile fever in Algeria [26]. The application of the Mahalanobis distance statistic to WNV distribution in North Africa underlined the fact that the suitable areas for WNV disease occurrence between May and July are located in Tunisia, Libya and Egypt. Conversely, during the summer/autumn months, this model characterized suitability for WNV disease in Morocco, north Tunisia and the Mediterranean coast of Africa. The persistence of suitable conditions in December is confined to the coastal areas of Morocco, Tunisia, Libya and Egypt [155]. These observations provide an interesting tool for a better surveillance planning of WNV occurrences in both humans and animals. While several vaccine candidates are available for horses and birds, no vaccine against WNV is currently licensed for human purposes [156]. However, some promising attempts are in progress on live attenuated vaccine [157]. Two recent studies carried out in Morocco reported, for the first time, the detection of WNV from *Cx. pipiens* mosquitoes and its circulation among horses, as well as the evidence of human and domestic birds infection, suggesting an active circulation of the virus [158,159].

#### 6.2.2. Rift Valley Fever (RVF)

RVFV (*Bunyaviridae*, *Phlebovirus*) is a zoonotic disease causing severe pathology in humans and livestock [160]. In ruminants it causes fever, weakness, anorexia and abortion, including mortality. Infection in humans may occur through infected mosquito bites, although the principal infection mode is through contact with infected animals and animal tissues (e.g., during slaughter) or products (raw milk). The human symptoms vary from mild clinical signs (flu-like symptoms) to death [87]. Studies conducted in the North Africa region using multicriteria decision analysis (MCDA), identified the Maghreb region as moderately suitable for the occurrence of RVF enzootics and highly suitable for RVF epizootics [87,161].

Human cases of RVF were reported at the southern borders of the Maghreb region identified as cases of virus introduction [76], as border countries like Mali and Mauritania are known to be endemic of RVF [9,10]. To date, no human cases have been reported in Algeria, Morocco or Tunisia [76]. However, a serological study conducted in Libya to determine the seroprevalence of zoonotic viruses and bacteria causing acute febrile illness revealed that 0.4% of 950 human blood samples tested have antibodies against RVFV [76]. In another serological study, antibodies specific for RVFV were not detected in any of the 857 collected samples among Libyan domestic ruminants, excluding the circulation of RVFV in the period 2015–2016 [162]. However, in Egypt the last epidemic occurred in 2003, resulting in human and animal infections [163]. Given the culture of the region and the religious practices, the Eid al-Adha holiday represents a period of public health risk. This event involves intensive movement, trade in sheep, and the slaughter of animals at home [87]. Health authorities should increase their vigilance during this period.

#### 6.2.3. Chikungunya Virus (CHIKV)

Chikungunya (*Togaviridae*, *Alphavirus*) derives from the Makondé language (Tanzania) and means ‘curling up’. The disease causes high fever, joint damage, headache, muscle pain and rash. Severe neurological forms, in exceptional cases, can be observed [65].

The mechanism of transmission is direct spillover, when an enzootic or bridge vector transmits the virus from an enzootic host to a human, or through human-to-human transmission [1]. Not yet detected in North Africa countries, its epidemiology is not well characterized, and its transmission is currently under-recognized [76,164]. In Egypt, human antibodies against CHIKV antigens were reported in an earlier study dating from 1985 [165].

#### 6.2.4. Dengue

Dengue is classified as a neglected tropical disease by the WHO, affecting over 120 countries. Currently, 70% of the burden is on the Asian continent. There are four DENV (*Flaviviridae*, *Flavivirus*) serotypes (DENV-1, DENV-2, DENV-3 and DENV-4) that can co-circulate within a region [166]. Dengue causes a wide spectrum of clinical manifestations, which can range from asymptomatic or mild illness, to severe flu-like symptoms. Less commonly, it can develop potentially lethal complications and become severe dengue [166]. The epidemiology of DENV in the North Africa region remains largely uncharacterized [167]. Only a few imported cases of dengue were reported between 2015 and 2016 in Algeria, and it has not yet been detected in Morocco [76].

Dengue has been more studied in Egypt, with sentinel reports of autochthonous transmission (2010) [167]. DENV transmission was suggested some years prior to this, in a report of two travelers diagnosed with dengue after returning from southern Egypt in 2011 [167]. After a break of ten years without a human dengue case, an outbreak occurred in Egypt [167]. Dengue fever has re-emerged, in the Assiut governorate in 2015 and Red Sea governorate in 2017 [168]. Several vaccines have been developed, and are currently going through the preclinical or clinical trial phases [169].

#### 6.2.5. Yellow Fever (YF)

Yellow fever (*Flaviviridae*, *Flavivirus*) is an acute viral hemorrhagic disease transmitted by *Aedes* and *Haemogogus* species [170]. It is distributed in tropical and subtropical areas of central and south America and Africa [170]. The virus was isolated for the first time in 1927, in a male patient [171]. YF infection can cause the onset of varying clinical features, ranging from a self-limited or mild febrile illness with flu-like symptoms in most cases, to severe hemorrhage and liver disease [171]. The “yellow” comes from jaundice, which affects some patients [170]. The YF vaccine is the best way means of preventing the disease, and is recommended for travelers to endemic areas [172]. Nowadays, no cases are reported in North Africa countries.

#### 6.2.6. Zika Virus

Zika (*Flaviviridae*, *Flavivirus*) is a cause of neurological disorders/neonatal malformations such as Guillain-Barré syndrome and fetal microcephaly [1]. Although sexual transmission has been reported, probably due to infection of the semen, sperm, and testicles [1], mosquitoes are the primary vector. For instance, there is evidence that bats may play a role in the ZIKV infectious cycle [173]. Zika virus is still unreported to date in the Maghreb [76]. Currently, as part of the “One Health” approach, MediLabSecure is conducting comprehensive risk assessments, establishing effective monitoring programs, developing preparedness activities, and implementing adequate control measures with regard to the Zika virus and other arboviral threats in the Euro-Mediterranean area, including Algeria, Egypt, Libya, Morocco and Tunisia [173].

#### 6.2.7. Sindbis Virus (SINV)

Human infections by the Sindbis virus (*Togaviridae*, *Alphavirus*) cause symptoms that include febrile illness, fever, arthritis, and maculopapular rash [174]. A serological study conducted in Libya to determine the seroprevalence of zoonotic viruses and bacteria causing acute febrile illness, revealed that 0.5% of 950 human blood samples tested had antibodies against SINV antigens [76]. It has been isolated from mosquitoes in the Nile Valley of Egypt [175]. Very recently, it was detected and isolated from *Cx. pipiens* and *Cx. perexiguus* collected in Timimoun, southern Algeria [176].

#### 6.2.8. Usutu Virus (USUV)

USUV (*Flaviviridae*, *Flavivirus*) is an emerging virus belonging to the JEV complex. In Tunisia, the first evidence of its circulation among equids was confirmed by the presence of neutralizing antibodies in 3.5% of tested equids, concomitant with WNV, by a serological study in the Kébili governorate in the southwest of the country [151]. Its circulation in Algeria, Morocco, Libya, and Egypt has not been yet reported, but it seems probable that USUV is locally present [177].

### 6.3. Bacteria

The involvement of mosquitoes in the transmission of bacterial pathogens to vertebrates has not been formally reported. However, in recent years, several studies have demonstrated the molecular detection of *R. felis* in *An. gambiae* and *Ae. albopictus* [84,178], as well as *R. felis* and *R. bellii* in *Cx*. *pipiens pallens* and *An*. *sinens* in mosquitoes [179]. Recently, an experimental model demonstrated the vector role of *An. gambiae* for *R. felis*, suggesting its involvement in the transmission of this pathogenic bacteria [180]. In a recent study, Zhang et al. observed seasonal and gender differences of *R. felis* detection and blood meal in field-collected mosquitoes. They therefore suggested the possible transstadial and transovarial transmission of *R. felis* in mosquitoes [181]. Furthermore, the role of mosquitoes as mechanical vectors of *Francisella* (*F.*) *tularensis* causing tularemia in humans was suggested.

DNA of *F. tularensis* subsp. *holarctica* was detected in both immature and adult stages of *Aedes* spp. collected in Örebro, a Swedish area [182]. Bäckman et al. demonstrated the possibility of the transmission of *F. tularensis* subsp. *holarctica* by adult *Ae. aegypti* having acquired the pathogen from their aquatic larval habitats [183]. The role of mosquitoes in the transmission of Lyme Borreliosis was also evoked. Spirochetes were detected in 1.7% of mosquito females, 3.2% of larvae and in 1.6% of pupae of *Cx. p. pipiens* sampled in the forested areas of Szczecin, Poland [184]. Spirochetes of *Borrelia* spp. have been detected in *Culex* spp., *Aedes* spp. and *Anopheles* spp. mosquitoes in Germany [185]. Very recently, Rudolf et al. provided the first mass molecular screening of *Ae. vexans*, *Cx. pipiens* and species of the *An. maculipennis* complex, collected in the Czech Republic and Slovakia, for *Bartonella* spp. The initial screening yielded 0.1%, 0.3% and 27.2% *Bartonella*-positive for the *A. vexans*, *C. pipiens* and *A. maculipennis* complex, respectively [186]. Obviously, molecular detection does not confirm the vector role, and further investigations are necessary to assess the mosquito’s role in the transmission of pathogenic bacteria species. Up to this date, all investigations into the detection of bacteria in the vector mosquito species mentioned above have been carried out in Europe, whereas no studies have been conducted in the countries of North Africa.

The circulation and distribution of MBDs mentioned above are still not well documented in the North Africa region, and more attention is recommended to prepare for possible epidemic outbreaks.

## 7. Mosquito Control Strategies in North Africa

Ingenious techniques are now used in the control of mosquitoes and surveillance of their transmitted diseases. The absence of effective vaccines and specific antiviral treatments for many of these arboviruses constrains researchers, who need to develop new approaches to control these diseases and their transmission. In North Africa, mosquito control programs include mosquito fauna surveillance, source reduction of populations, and a variety of control strategies targeting larval and adult mosquitoes. Interventions against mosquito populations occur either to prevent pathogen transmission or to reduce pests generated by the installation of invasive species such as *Ae. albopictus*. In Algeria, mosquito control strategies target mainly the immature stages. The most frequent schedules consist of the suppression or modification of aquatic habitats by the application of chemical reagents (i.e., insecticides) wherever the removal of stagnant water sites is not possible, such as in the flooded cellars of buildings.

Nevertheless, public education remains the most efficient way to reduce the nuisance induced by mosquitoes and therefore MBDs. Sensitization could be carried out by creating advertising spots in the media and on social networks, using popular scientific content accessible to the general public, as well as in schools. North Africa countries should move towards integrated vector management (IVM) [187]. In this context, *Wolbachia* was intensively studied as a promising agent to control diseases transmitted by mosquitoes. In Egypt, a study conducted in the Assiut governorate aimed to simultaneously detect *Wolbachia* and filarial worms in mosquitoes. Among the tested mosquitoes, 0.24% were positive for *Wolbachia*; it was also observed that *Wolbachia* altered *Culex* spp. as a primary vector for *W. bancrofti*, being replaced by *Anopheles* sp. in this region [135]. Atyame et al., conducted a study to map the distribution areas of *Wolbachia* strains in *Cx. pipiens* (*w*Pip) in Algeria and Tunisia. They characterized their cytoplasmic incompatibility (CI) patterns from isofemale lines sampled in different localities, and analyzed CI expression from direct observations in the field [188]. In Morocco, the diversity of *w*Pip in natural populations of two forms of *Cx. pipiens* (*pipiens* and *molestus* forms) was tested in three cities of Morocco. The authors found three types, I, IV and V, detected for the first time in North Africa [189].

### Insecticide Resistance

Resistance of mosquitoes to insecticides is a worrisome concern in African countries, especially in areas where malaria transmission occurs. Accumulative data are available on insecticide resistance in the mosquito species of North Africa. However, few studies have been conducted.

A study conducted on *Cx. p. pipiens* mosquitoes collected in northern Algeria identified two mechanisms responsible for resistance to organophosphate insecticides: (i) overproduction of detoxifying esterases, encoded by the ester superlocus, and (ii) insensitivity to acetylcholine esterase, encoded by the ace-1 gene [190]. A study of insecticide resistance was conducted in the center of Morocco (Fez locality). It aimed to test *Cx. pipiens* larvae and adults for susceptibility or resistance to temephos insecticide using WHO sensitivity tests. The bioassay results revealed the presence of resistance in *Cx. pipiens* larvae to temephos [191]. A second study was recently conducted in three sites (Tangier, Casablanca and Marrakech), which aimed to test for lambda-cyhalothrin resistance and to detect L1014F kdr mutation in *Cx. pipiens* forms. The results indicated that 21% of the tested population was resistant to lambda-cyhalothrin and that the *pipiens* form was more resistant than the *molestus* form.

L1014F kdr mutation was detected in different forms of *Cx. pipiens* from different sites [192]. In Tunisia, Tabbabi et al. conducted several studies on resistance status to pyrethroids, organophosphate and temephos insecticides in *Cx. pipiens*, *An. labranchiae* and *An. sergentii* either in larvae or adult specimens [193,194,195,196]. The susceptibility status of *Cx. p. pipiens* populations against deltamehtrin insecticide ranged from 0.67 to 31.4 [196]. Results revealed that the distribution of resistance ratios to organophosphate in *Cx. p. pipiens* appeared to be influenced by the degree of urbanization [195]. The same species exhibited tolerance to temephos with low and high levels of resistance [193]. In another study, *An. labranchiae* showed low and medium resistance ratios to temephos insecticide [194]. In Morocco, Arich et al. found that *Cx. pipiens* forms *pipiens* and *molestus* were resistant to most common insecticide families used in mosquito control as well as in agriculture or urban residuals, all over Morocco [197]. Further studies are needed to elucidate the reality of insecticide resistance of mosquito vector species in North Africa, in order to promote integrated control strategies.

## 8. Potential Factors Contributing to the Future Spread of MBDs within the North Africa Region

Countries of North Africa have southern borders with Mauritania, Mali, Niger, Chad and Sudan where diseases such as malaria, CHKV and RVFV are endemic. Indeed, recent reports (September 2021) state cases of RVFV in Algeria and Libya among livestock [198]. Malaria cases have been reported in Algeria, one year after being declared malaria-free by the WHO in 2019. A thousand cases have been declared in the extreme south of the country, affecting several provinces. The investigation revealed that cases were all imported by nomads, who crossed the borders even if they were closed due to the COVID-19 pandemic.

Circulation within the North Africa region is also observed, particularly land transportation between Algeria and Tunisia. In addition, of North Africa have a lot of human and commercial exchanges, especially transportation via seaports, with the countries of the north coast-line of the Mediterranean such as France, Spain and Italy. Indeed, these European countries observed outbreaks of CHIKV, DENV, and ZIKV, shortly after the establishment of the *Ae. albopictus.* North Africa region is at risk of undergoing the same scenarios, especially if the *Ae. albopictus* populations are competent for CHIKV, DENV, and ZIKV transmission, as was evaluated in Tunisia [199]. In addition to *Ae. albopictus*, other invasive species such as *Ae. japonicus* (Japanese mosquito), *Ae. koreicus* (Korean mosquito) and *Ae. Atropalpus,* which have already been reported in Europe, are also a threat to North Africa countries, and may participate in the emergence and rapid expansion of CHIKV, DENV, and ZIKV. The surveillance of points of entry such as airports and ports, is of crucial importance [200]. In addition, other global factors, such as climate change, rapid population growth, deforestation associated with uncontrolled urbanization, and the emergence of mosquitoes resistant to common insecticides, can strongly contribute to the expansion of MBDs in North Africa [201].

## 9. Conclusions

The present review provides information on the vector mosquito species present in North Africa, and their associated diseases. Twenty-six mosquito species stand out as pathogen vectors, including the *Anopheles* (12 species), *Aedes* (five species), *Culex* (eight species) and *Culiseta* (one species) genera. Overall, seven MBDs may occur in North Africa, two parasitic infections (malaria and filariasis) and five viral infections (WNV, RVF, DENV, SINV and USUV). Recent scientific evidence confirms the introduction of *Ae. albopictus* in several countries from North Africa. Since then, the risk of emergence of CHIKV and ZIKV with autochthonous cases remains significant, notably in densely populated cities in Algeria, Morocco and Tunisia. Entomological studies conducted recently in North Africa countries confirm the persistence of vectors or potential vector species, corroborating the fact that the risk of outbreaks is highly probable. The insufficient scientific work, notably in the field of entomology, is an additional factor which could impede the monitoring and prevention of MBDs. Further studies on mosquito vectors in North Africa are required to complete the epidemiological picture of this region of the world. Collaboration between these North Africa countries would certainly improve the knowledge of the dynamics of viral circulation and other aspects of MBDs in the region. In this way, the establishment of an integrated system for arboviruses vector control management on the model of current international collaborative malaria control programs appears compulsory.

## Figures and Tables

**Table 1 insects-13-00962-t001:** Main characteristics of currently available mosquito vector species genomes.

Species	Diseases	Genome Size (Mb)	G + C (%)	Protein Coding Genes	GenBank Genome ID	Geographic Distribution	Reference
*Anopheles arabiensis*	Malaria	256.8	44.8	25,532	11,544	Egypt.	[41]
*Anopheles coluzzi*	Malaria	273.4	44.5	23,396	41,035	Algeria, Morocco, Tunisia.	[41]
*Anopheles stephensi*	Malaria	221	44.8	11,789	2653	Egypt.	[42]
*Aedes albopictus*	Arboviruses Filariasis	1967	40.4	17,539	45	Algeria, Morocco, Tunisia.	[43]
*Aedes aegypti*	Arboviruses	1376	38.2	15,419	44	Algeria, Morocco, Egypt, Libya, Tunisia.	[44]
*Culex quinquefasciatus*	Arboviruses Lymphatic filariasis	579	37.4	18,883	393	Algeria, Morocco.	[45]

**Table 2 insects-13-00962-t002:** List of the *Culicidae* species that commonly act as vectors or potential vectors in human and animal diseases and their distribution in North Africa.

Subfamily	Genus	Subgenus	Species	Diseases Transmitted			Distribution	References
				Viruses	Parasites	Bacteria		
** *Culicinae* **	** *Aedes* **	*Stegomyia*	*Ae. albopictus*	Dengue, Yellow fever **, Chikungunya **, Zika	Filariasis	*Rickettsia felis*	Algeria Morocco	[54,55]
			*Ae. aegypti*	Dengue, Yellow fever **, Chikungunya **, Zika,	Filariasis	/	Algeria, Egypt, Libya, Morocco, Tunisia.	[56]
		*Ochlerotatus*	*Ae. caspius*	Rift Valley Fever, West Nile Virus, Usutu Virus	Filariasis	/	Algeria, Egypt, Libya, Morocco, Tunisia.	[57,58]
			*Ae. detritus*	Rift Valley Fever, WNV	/	/	Algeria, Egypt, Tunisia	[57,58]
		*Aedimorphus*	*Ae. vexans*	Rift Valley Fever,	Filariasis	/	Algeria, Morocco	[59,60]
	** *Culex* **	*Culex*	*Cx. pipiens*	West Nile Virus, Rift Valley Fever Virus, Usutu Virus,	*Dirofilaria immitis, * *D. repens, Wuchereria bancrofti*	/	Algeria, Egypt, Libya, Morocco, Tunisia	[57,58,59]
			*Cx. antennatus*	West Nile Virus, Rift Valley Fever Virus	*Dirofilaria repens*	/	Algeria, Egypt, Tunisia	[61,58]
			*Cx. perexiguus*	Rift Valley Fever Virus, West Nile Virus, Sindbis virus	/	/	Algeria, Egypt, Libya, Morocco, Tunisia.	[57,58]
			*Cx. theileri*	Rift Valley Fever Virus, West Nile Virus, Sindbis virus	/	/	Algeria, Egypt, Libya, Morocco, Tunisia.	[57,58]
			*Cx. univittatus*	West Nile Virus, Sindbis virus	/	/	Egypt	[58]
			*Cx. quinquefasciatus*	West Nile Virus, Saint Louis encephalitis **	*Wuchereria bancrofti*	/	Algeria, Morocco.	[59,62] ^1^
		*Barraudius*	*Cx. modestus*	WNV, Usutu virus, Tahyna virus **, Lednice virus **	/	/	Algeria, Morocco	[63]
			*Cx. pusillus*	/	*Dirofilaria immitis*	/	Algeria, Egypt, Libya, Tunisia.	[61]
	** *Culiseta* **	*Culiseta*	*Cs. annulata*	Tahyna virus **	/	/	Algeria, Morocco, Tunisia.	[21]
** *Anophelinae* **	** *Anopheles* **	*Anopheles*	*An. labranchiae*	/	*Plasmodium*	/	Algeria, Libya Morocco, Tunisia.	[64]
			*An. algeriensis*	/	*Plasmodium*	/	Algeria, Egypt, Libya, Morocco, Tunisia.	[64,58]
			*An. coustani*	Rift Valley Fever Virus, Zika	*Plasmodium*	/	Egypt	[58]
			*An. claviger*	/	*Plasmodium*	/	Algeria, Libya, Morocco, Tunisia.	[18,65,66,67]
			*An. sachorovi*	/	*Plasmodium*	/	Algeria	[68]
		*Cellia*	*An. multicolor*	/	*Plasmodium*	/	Algeria, Egypt, Libya, Morocco, Tunisia.	[64,58]
			*An. coluzzi*	O’nyong-nyong ** Tataguine virus **, Nyando virus **	*Plasmodium*	*Rickettsia felis*	Algeria *, Egypt #	[69,70]
			*An. sergentii*	/	*Plasmodium*	/	Algeria, Egypt, Libya, Morocco, Tunisia.	[64,58]
			*An. superpictus*	/	*Plasmodium*	/	Algeria, Egypt	[17,58]
			*An. pharoensis*	/	*Plasmodium * *Filariasis*	/	Egypt	[58]
			*An. stephensi*	/	*Plasmodium*	/	Egypt	[58]
			*An. arabiensis*	/	*Plasmodium*	/	Egypt	[18]

* Occasional detection, **** Not reported yet, # Formerly, ^1^ hybrid *Cx. pipiens*/*Cx. quinquefasciatus*.

## Data Availability

Not applicable.

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
