# Peer review of "Mosquito Vectors (Diptera: *Culicidae*) and Mosquito-Borne Diseases in North Africa"

_insects, 2022, doi:10.3390/insects13100962_

Round 1
Reviewer 1 Report (Previous Reviewer 2)
The manuscript is much improved. I have no further suggestions
Author Response
Reviewer #1
- The manuscript is much improved. I have no further suggestions
Response: We thank the reviewers for all the remarks and suggestions that have been issued before to improve the quality of the present manuscript.
Reviewer 2 Report (New Reviewer)
This review is about 26 mosquito species and mosquito-borne diseases in North Africa. Authors performed a good job, however they need to consider some points.
Simple summary: Please mention the number of mosquitoes species and mosquito-borne diseases you confirmed in this study. So that reader will have idea about the depth of work.
Abstract: Authors should write the results of their findings after reading and analysis of published literature. Provide some detail of mosquito-borne diseases, how many? major mosquito-borne diseases/highly infectious in the region etc. How many parasitic, bacterial and how many viral diseases are prevalent in the region? Abstract gives the impression of more introduction. Detailed abstract will intend reader to explore more. Abstract is poorly written, sorry.
Introduction: It is ok.
Methodology: Methodology is missing. Authors are not saying it systematic review, however, they should provide the some details how they extracted the literature used in the study. On which criteria they selected papers to include in the study or exclude. How many papers they included or excluded and how many publications were total? How you removed duplicates?
Line 78-85: passage they can move under heading of methodology.
Line 82: Social networking font is different.
Provide full detail about keywords used to search literature.
Which timeline authors selected to search the literature e.g if the review of literature is from 1947-2022 etc. This is an example. Author should provide timeline of the this review.
Results and discussion: After methodology, authors can use heading for results and discussion. They can use subheadings onward as 3.1. Salient data on North Africa and continue......
Morphology, life cycle, taxonomy, genomics and identification methods of Culicidae:
Please summarize the identification methods used in North African region. The literature used under this heading is global (35-38).
Please proofread carefully to correct all English language errors.
Conclusion is nicely written.
Author Response
Reviewer #2
General comment.
- This review is about 26 mosquito species and mosquito-borne diseases in North Africa. Authors performed a good job, however they need to consider some points.
Specific comments.
- Simple summary: Please mention the number of mosquitoes species and mosquito-borne diseases you confirmed in this study. So that reader will have idea about the depth of work.
Response: We thank the reviewer for the suggestion. This information is now added to the simple summary.
- Lines 28-30: In this review, we focused only on mosquitoes of medical and veterinary importance (26 species), providing a concise update of the literature on species and diseases they carried (two parasitic and five viral infections).
- Abstract: Authors should write the results of their findings after reading and analysis of published literature. Provide some detail of mosquito-borne diseases, how many? major mosquito-borne diseases/highly infectious in the region etc. How many parasitic, bacterial and how many viral diseases are prevalent in the region? Abstract gives the impression of more introduction. Detailed abstract will intend reader to explore more. Abstract is poorly written, sorry.
Response: As recommended by the reviewer #2, details were added to the abstract about parasitic and viral infections as follow:
Lines 44-46: These 26 species are involved in the circulation of seven MBDs in North Africa, including two parasitic infections (malaria and filariasis) and five viral infections (WNV, RVF, DENV, SINV and USUV). No bacterial diseases have been reported so far in this area.
- Introduction: It is ok.
Response: We thank the reviewer for this comment.
- Methodology: Methodology is missing. Authors are not saying it systematic review, however, they should provide the some details how they extracted the literature used in the study. On which criteria they selected papers to include in the study or exclude. How many papers they included or excluded and how many publications were total? How you removed duplicates?
Response: The reviewer is right, a section under the introduction untitled”2. Methodology” (line 100) was added to the manuscript. In addition to the description of the strategy applied to retrieve the most relevant articles and publications which was already mentioned at the end of the introduction, other details on the number of publication collected and included after applying excluding criteria were presented, as suggested by the reviewer.
Lines 108-110: More than 300 publications have been recovered using the keywords above focusing essentially on the last two decades, and then around 100 were excluded either because the article is not fully accessible, or to avoid redundancies.
- Line 78-85: passage they can move under heading of methodology.
Response: This passage is now moved under the heading of methodology, as recommended by the reviewer.
- Line 82: Social networking font is different.
Response: We thank the reviewer for this remark and the type of font was homogenized in this line and throughout the manuscript.
- Provide full detail about keywords used to search literature.
Response: Keywords used to search literaturewere already mentioned in the manuscript (line 102). No changes have been made.
- Which timeline authors selected to search the literature e.g if the review of literature is from 1947-2022 etc. This is an example. Author should provide timeline of the this review.
Response: The reviewer is right; in this review authors provide an update focusing on advances in mosquitoes and the diseases they transmit over the past two decades. This detail is now added in the Methodology section.
Line 108: More than 300 publications have been recovered using the keywords above focusing essentially on the last two decades, …
- Results and discussion: After methodology, authors can use heading for results and discussion. They can use subheadings onward as 3.1. Salient data on North Africa and continue......
Response: By adding a section named 2. Methodology, the titles and subtitles that come after are now arranged accordingly proposed by the reviewer.
- Morphology, life cycle, taxonomy, genomics and identification methods of Culicidae:
Please summarize the identification methods used in North African region. The literature used under this heading is global (35-38).
Response: We thank the reviewer for this remark. However, as mentioned in the manuscript lines 152-154 “Most of studies conducted in North Africa on mosquito species used morphological keys; it stays the standard method to identify collected species.”In such condition, to our own opinion, we consider that this section is essential for this review and should be maintain notably in order to inform entomologists (students and researchers) about the recent advances for mosquito identification.
- Please proofread carefully to correct all English language errors.
Response: As recommended by the reviewer, the manuscript was double checked and English errors were corrected.
- Conclusion is nicely written.
Response: We thank the reviewer; the quality of the manuscript has been greatly improved thanks to the comments of the reviewers.
This manuscript is a resubmission of an earlier submission. The following is a list of the peer review reports and author responses from that submission.
Round 1
Reviewer 1 Report
This paper aimed to update the distribution of mosquitoes vector species, and the status of infectious diseases they transmit in North Africa. For it, the authors provided a comprehensive literature review. This study is interesting and important in its field. However, some adjustment should be made to improve the final version of this paper. I have some comments/suggestions:
L. 27-29 “Overall, 26 mosquito species presumed or confirmed as pathogenic agent vectors, including 5 Aedes spp., 8 Culex spp., 1 Culiseta spp. and 12 Anopheles spp. occur currently in North Africa. To facilitate easier reading, I suggest: Currently, 26 mosquito species confirmed as pathogenic vectors occur in North Africa, including Aedes (5 species), Culex (8 species), Culiseta (1 species) and Anopheles (12 species)
L. 107 - 3. Morphology, life cycle, taxonomy and genomics of Culicidae and L. 685 - 7. Identification methods of mosquitoes are connected. Therefore, I suggest to present these two subsection in one.
Table 2 – Aedes spp., Culex spp., Culiseta spp., Anopheles spp., “spp.” are not essential on genus columns, please remove them.
Why is there an interrogation mark in Rickettsia felis, and in Parasites column of An. coustani? Please explain this on the table’s footnote.
Please add the subgenus/subgenera of Anopheles species.
A space is missing in some mosquitoes species (Cx.pipiens, Cx.perexiguus, An.gambiae, An.algeriensis, An.sergentii, An.superpictus). Please correct it.
Please, review all the scientific names, and make sure they are spelled correctly. Some of them require parentheses, others do not (e.g. L.285 (Felt) does not requires parentheses). Do not forget to add a comma between the authority and the year of publication; and also review the main text, there are many words with missing space between them (e.g. L. 144, Table2; L. 248, efficientvectors; L. 423, D. immitisantigens, and many more).
L. 139, 209, 285, 294 do not require spp., please remove them
L.155 Please correct Ae. (Stegomyia) albopictus (Skuse, 1894) to (Skuse, 1895)
L. 180, 191, 197, 203, 212-215, 226, 288, 297-302, 309, 315, 322, 329, 350, 353, 357, 360, add a comma between the authority and the year of publication
L.209 Please correct Culex spp. (Linne) to Culex Linnaeus
In table 2, a list of mosquito vector species and the disease transmitted by them are presented. In L. 368 the mosquito-borne diseases in North Africa are presented. Although no cases of ONNV (L. 571) have been reported in North Africa, a short paragraph with some information is presented. Even though it does not occur in Nrth Africa, it would be interesting to add some information about the other arboviruses mentioned in table 2 (yellow fever, Saint Louis encephalitis, Tahyna virus, Lednice virus, Tataguine Nyando).
L. 370 Correct Plasmodiums to Plasmodium
L.371 “The hematozoa responsible of malaria was discovered at Constantine, Algeria, in 1880, by the Dr Alphonse Laveran. Four years later (ie, 1884), he published the vector role of the mosquito in the spread of the disease”. Please rewrite this, correct “of malaria” to “for malaria”, “in the spread of the disease” to “in the transmission of the disease”, remove “the” before Dr., remove (ie 1884) – this is not essential.
L.399, 413 scientific names should be in italic.
L. 406, 408 Correct Wu. Boncrofti and Wu. Bancrofti to Wu. bancrofti
L.622 Correct recommanded to recommended
L. 725 Correct Ae. Japonicas to Ae. japonicus
In the Conclusion, I did not see the findings on MBDs. I suggest to rephrase your research topic so that it is clearer to the reader (the distribution on mosquitoes species vectors (how many species? New species reported? Emergence risk?), and the status of infectious diseases in North Africa (How many MBDs? xx parasitic infections, xx arboviruses, and xx bacterial infection occur in North Africa, Recent transmission? Emergence risks?); state the significance (e.g. introduction of Ae. albopictus, and other invasive species (Ae. japonicus, An. koreicus); and remind the reader why the work presented in this paper is relevant (lack of vector control and MDBs surveillance in North Africa, and the need for further research on this field).
Author Response
Reviewer #1
This paper aimed to update the distribution of mosquitoes vector species, and the status of infectious diseases they transmit in North Africa. For it, the authors provided a comprehensive literature review. This study is interesting and important in its field. However, some adjustment should be made to improve the final version of this paper. I have some comments/suggestions:
- L. 27-29 “Overall, 26 mosquito species presumed or confirmed as pathogenic agent vectors, including 5 Aedes spp., 8 Culex spp., 1 Culiseta spp. and 12 Anopheles spp. occur currently in North Africa. To facilitate easier reading, I suggest: Currently, 26 mosquito species confirmed as pathogenic vectors occur in North Africa, including Aedes (5 species), Culex (8 species), Culiseta (1 species) and Anopheles (12 species)
Response: For easier reading the sentence has been modified, as recommended by the reviewer.
- Lines 42-44: Currently, 26 mosquito species confirmed as pathogen vectors occur in North Africa, including Aedes (5 species), Culex (8 species), Culiseta (1 species) and Anopheles (12 species). This review may guide research studies…
- L. 107 - 3. Morphology, life cycle, taxonomy and genomics of Culicidae and L. 685 - 7. Identification methods of mosquitoes are connected. Therefore, I suggest to present these two subsection in one.
Response: We thank the reviewer for the suggestion, as recommended, the two previous subsections: 3 (Morphology, life cycle, taxonomy and genomics of Culicidae) and 7 (Identification methods of mosquitoes), have been combined into a single section entitled:
- Morphology, life cycle, taxonomy, genomics and identification methods of Culicidae, in lines 123-163.
- Table 2 – Aedes spp., Culex spp., Culiseta spp., Anopheles spp., “spp.” are not essential on genus columns, please remove them.
- Response: As recommended by the reviewer, “spp.” was removed from genus columns of the table 2.
- Why is there an interrogation mark in Rickettsia felis, and in Parasites column of An. coustani? Please explain this on the table’s footnote.
- Response: We apologize for this error, the interrogation marks were removed from table 2.
- Please add the subgenus/subgenera of Anopheles species.
- Response: As recommended by the reviewer, the subgenera of Anopheles species were added on the table 2. For formatting purposes, table 2 has been slightly modified.
- A space is missing in some mosquitoes species (Cx.pipiens, Cx.perexiguus, An.gambiae, An.algeriensis, An.sergentii, An.superpictus). Please correct it.
Response: These points are now corrected.
- Please, review all the scientific names, and make sure they are spelled correctly. Some of them require parentheses, others do not (e.g. L.285 (Felt) does not requires parentheses). Do not forget to add a comma between the authority and the year of publication; and also review the main text, there are many words with missing space between them (e.g. L. 144, Table2; L. 248, efficientvectors; L. 423, D. immitisantigens, and many more).
Response: We apologize for these inconveniences, it may have happened, during the exchanges between the co-authors, who have different word versions. All these points are now corrected throughout the manuscript.
- L. 139, 209, 285, 294 do not require spp., please remove them
Response: spp. is now removed in lines 172, 250, 323 and 333.
- L.155 Please correct Ae. (Stegomyia) albopictus (Skuse, 1894) to (Skuse, 1895)
Response: This point is now corrected in line 188.
- L. 180, 191, 197, 203, 212-215, 226, 288, 297-302, 309, 315, 322, 329, 350, 353, 357, 360, add a comma between the authority and the year of publication
Response: A comma is now added in lines 214, 228, 235, 241, 266, 296, 304, 307, 313, 316, 319, 321, 323, 326, 348, 353, 360, 366, 381, 385, 388, 392 and 396.
- L.209 Please correct Culex spp. (Linne) to Culex Linnaeus
Response: This point is now corrected in line 250.
- In table 2, a list of mosquito vector species and the disease transmitted by them are presented. In L. 368 the mosquito-borne diseases in North Africa are presented. Although no cases of ONNV (L. 571) have been reported in North Africa, a short paragraph with some information is presented. Even though it does not occur in Nrth Africa, it would be interesting to add some information about the other arboviruses mentioned in table 2 (yellow fever, Saint Louis encephalitis, Tahyna virus, Lednice virus, TataguineNyando).
Response: We thank the reviewer for these suggestions. A section about yellow fever was added to the manuscript as it is a major arbovirus. The section about ONNV was deleted as no cases were reported in North Africa. No information has been included for Saint Louis encephalitis, tahyna virus, lednice virus, tataguine virus and nyando virus. However, a footnote (**) was added to the table 2 to indicate viruses that have not been detected in North Africa yet.
- Lines 588-597 : 5.2.5. Yellow fever (YF)
Yellow fever (Flaviviridae, Flavivirus) is an acute viral haemorrhagic disease transmitted by Aedes and Haemogogus species [161]. It is distributed in tropical and subtropical areas of Central and South America and Africa [161]. The virus was isolated for the first time in 1927 in a male patient [162]. YF infection can cause the onset of varying clinical features, ranging from a self-limited or mild febrile illness with flu-like symptoms in most of the cases to severe hemorrhage and liver disease [162]. The “yellow” comes from jaundice that affects some patients [161]. The YF vaccine is the best way means of preventing the disease, it is recommended for travelers to endemic areas [163]. Nowadays, no cases were reported in North African countries.
- L. 370 Correct Plasmodiums to Plasmodium
Response: As recommended by the reviewer, this point is now corrected in line 407.
- L.371 “The hematozoa responsible of malaria was discovered at Constantine, Algeria, in 1880, by the Dr Alphonse Laveran. Four years later (ie, 1884), he published the vector role of the mosquito in the spread of the disease”. Please rewrite this, correct “of malaria” to “for malaria”, “in the spread of the disease” to “in the transmission of the disease”, remove “the” before Dr., remove (ie 1884) – this is not essential.
Response: As recommended by the reviewer, all these points are now corrected.
- Lines 408-410: The hematozoa responsible for malaria was discovered at Constantine, Algeria, in 1880, by Dr Alphonse Laveran. Four years later, he published the vector role of the mosquito in the transmission of the disease [110].
- L.399, 413 scientific names should be in italic.
Response: Wuchereria bancrofti, Dirofilaria immitis and Dirofilaria repens have been italicized in the revised manuscript.
- L. 406, 408 Correct Wu. Boncrofti and Wu. Bancrofti to Wu. bancrofti
Response: These points are now corrected in lines 443 and 445.
- L.622 Correct recommanded to recommended
Response: This point is now corrected in the revised manuscript.
- 725 Correct Ae. Japonicas to Ae. Japonicus
Response: This point is now corrected the revised manuscript, line 783.
- In the Conclusion, I did not see the findings on MBDs. I suggest to rephrase your research topic so that it is clearer to the reader (the distribution on mosquitoes species vectors (how many species? New species reported? Emergence risk?), and the status of infectious diseases in North Africa (How many MBDs? xx parasitic infections, xx arboviruses, and xx bacterial infection occur in North Africa, Recent transmission? Emergence risks?); state the significance (e.g. introduction of Ae. albopictus, and other invasive species (Ae. japonicus, An. koreicus); and remind the reader why the work presented in this paper is relevant (lack of vector control and MDBs surveillance in North Africa, and the need for further research on this field).
Response: We thank the reviewer for these relevant remarks; the conclusion was rewritten following the recommendations.
- Lines 793-814:
The present review provides information on the vector mosquito species present in North Africa and their associated diseases. Twenty-six mosquito species stand out as pathogen vectors, including Anopheles (12 species), Aedes (5 species), Culex (8 species) and Culiseta (1 species) genera. Overall, 7 MBDs may occur in North Africa, 2 parasitic infections (malaria and filariasis) and 5 viral infections (WNV, RVF, DENV, SINV and USUV). Recent scientific evidence support the introduction of Ae. albopictus in several countries from North Africa. Since then, the risk of emergence of CHIKV and ZIKV with autochthonous cases remains significant, notably, in densely populated cities from Algeria, Morocco and Tunisia. Entomological studies conducted recently in North African countries confirm the persistence of vectors or potential vectors species, corroborating that the risk of outbreaks is highly probable. The insufficient scientific works, notably, in the field of entomology, are additional factors which could impede the monitoring and prevention of MBDs. Further studies on mosquito vectors in North Africa are required to complete the epidemiological picture of this region of the world. Collaboration between these North African countries would certainly improve the knowledge of the dynamics of viral circulation and other aspects of MBDs in the region. In this way, the establishment of an integrated system for arboviruses vector control management on the model of current international collaborative malaria control programs appears compulsatory.
Reviewer 2 Report
In this review manuscript, the authors aimed to summarize the current status of mosquito-borne disease transmission in North Africa. MBD in Africa is definitely a hot topic and I agree with the authors about the need to discuss this subject. This is, however, a very complex and wide subject, and is very difficult to concisely organize all the information in one review manuscript. This led to two main issues with this manuscript:
The oversimplification of important ecological processes like in the example below.
“Ae. aegypti is native to West Africa, originally it was a zoophilic mosquito living in African forest environments. Then, due to the anthropization of forest areas and the practice of water storage, mosquito populations have adapted to humans [39].”
39. Duvallet G, Fontenille D, Robert V. Entomologie médicale et vétérinaire. 2017.
Many other factors were important to the adaptation of Ae. aegypti (and other species) to urban environments. Furthermore, other and more appropriate references are available to support this sentence. This issue can be found in other sections and should be addressed before this manuscript is accepted for publication.
The lack of important information such as in sections 6, 7, and 8. These are, in my opinion, the most important sections of the manuscript as they shed light on what’s to come and key information is missing.
Section 6. Mosquito control strategies in North Africa only touch on Wolbachia and there are so many more (and more important) strategies that need to be discussed under the IVM framework (that was never mentioned).
Section 7. Identification methods of mosquitoes. Why is that needed? What species cannot be identified by traditional methods? Is it possible to use molecular biology to identify mosquitoes in surveillance programs across North Africa? How much would it cost and is it feasible?
Section 8. Potential factors contributing to the future spread of MBDs within the North African region barely discuss what are the potential factors and what are the current predictions for the spread of MDBs in North Africa.
Specific comments:
More information is needed about dengue. Please include all the serotypes.
Dengue and chikungunya should be written in low caps
Abbreviations should be standardized throughout the manuscript.
Double check the text as there are many typos in the manuscript and italicize all scientific names.
Please remove this section as it has nothing to do with mosquitoes - 5.2.9. SARS-COV-2.
Author Response
Reviewer #2
In this review manuscript, the authors aimed to summarize the current status of mosquito-borne disease transmission in North Africa. MBD in Africa is definitely a hot topic and I agree with the authors about the need to discuss this subject. This is, however, a very complex and wide subject, and is very difficult to concisely organize all the information in one review manuscript. This led to two main issues with this manuscript:
- The oversimplification of important ecological processes like in the example below.
“Ae. aegypti is native to West Africa, originally it was a zoophilic mosquito living in African forest environments. Then, due to the anthropization of forest areas and the practice of water storage, mosquito populations have adapted to humans [39].”
- Duvallet G, Fontenille D, Robert V. Entomologie médicale et vétérinaire. 2017.
Many other factors were important to the adaptation of Ae. aegypti (and other species) to urban environments. Furthermore, other and more appropriate references are available to support this sentence. This issue can be found in other sections and should be addressed before this manuscript is accepted for publication.
Response: We thank the reviewer for this relevant remark, as it was recommended ecological factors contributing to the adaptation of Ae. Aegypti and its rapid worldwide invasion were cited.
- Lines 218-221: Then, several factors have contributed to its adaptation to the urban environment and worldwide invasion, in particular: the genetic predisposition, drought-resistant eggs, preferences for human blood, larval development in man-made container habitats [66].
- The lack of important information such as in sections 6, 7, and 8. These are, in my opinion, the most important sections of the manuscript as they shed light on what’s to come and key information is missing.
Response: We thank the reviewer for these suggestions which improve manuscript quality. More information has been added for the cited sections, see below.
- Section 6. Mosquito control strategies in North Africa only touch on Wolbachia and there are so many more (and more important) strategies that need to be discussed under the IVM framework (that was never mentioned).
Response: Integrated vector management framework was now mentioned in the manuscript (line 699). However, only few sentences refer to this point because the goal of the present review is to present what was done in North Africa, up to now. Moreover, to our knowledge and bibliographic researches, IVM is not yet installed in North Africa. Nevertheless, we are completely in agreement with the reviewer on the importance of IVM, that why a sentence was also added in the conclusion section as a recommendation.
- Lines 698-699: North African countries should move towards integrated vector management (IVM) .
- Section 7. Identification methods of mosquitoes. Why is that needed? What species cannot be identified by traditional methods? Is it possible to use molecular biology to identify mosquitoes in surveillance programs across North Africa? How much would it cost and is it feasible?
Response: As mentioned in the manuscript lines 144-155: Identification of specimens at species level is particularly essential to know exactly which mosquito species are involved in a transmission cycle of a vector disease. This section was included in order to inform entomologists about the recent advances for mosquito identification with relevant tools and low costs. Conversely to conventional methods like morphology identification which required entomological expertise or molecular tools which involve relative elevate coast, emerging methods for mosquito identification necessitating less financial and expertise resources appears as a reliable alternative. The application of molecular biology for mosquito fauna determination in a surveillance program at the country scale could be rapidly prohibitive. That why, as underlined in the end of this section, now combined with the “Morphology, life cycle, taxonomy, genomics of Culicidae” part, a mix of several methods, complementary for mosquito identification, seems more judicious in the frame of an IVM framework, as suggested by Schaffner [40].
- Section 8. Potential factors contributing to the future spread of MBDs within the North African region barely discuss what are the potential factors and what are the current predictions for the spread of MDBs in North Africa.
Response: The reviewer is right. In addition to those already mentioned in the manuscript, other potential factors contributing to the spread of MBDs in North Africa were added.
- Lines 787-790: In addition, other global factors, such as climate changes, rapid population growth, deforestation associated with uncontrolled urbanization and the emergence of mosquitoes resistant to common insecticides, can strongly contribute to the expansion of MBDs in North Africa [192].
- Specific comments:
- More information is needed about dengue. Please include all the serotypes.
Response: As recommended by the reviewer, section about dengue has been enriched with more information and dengue serotypes were included.
- Lines 571-576: Dengue is classified as a neglected tropical disease by the WHO, affecting over 120 countries. Currently, 70% of the burden is in Asian cotinent. There are four DENV (Flaviviridae, Flavivirus) serotypes (DENV-1, DENV-2, DENV-3 and DENV-4) that can co-circulate within a region [157]. Dengue causes a wide spectrum of clinical manifestations, it can range from asymptomatic or mild illness, to a severe flu-like symptoms. Less common, it can develops into a potentially lethal complication, named severe dengue [157]. The epidemiology…
- Lines 585-586: Several vaccines have been developed and are currently going through the preclinical or clinical trial phases [160].
- Dengue and chikungunya should be written in low caps
Response: Dengue, chikungunya and zika were now written in lowercase, except at the beginning of the sentence.
- Abbreviations should be standardized throughout the manuscript.
Response: Abbreviations are now standardized throughout the manuscript.
- Double check the text as there are many typos in the manuscript and italicize all scientific names.
Response: As recommended by the reviewer, the manuscript was double checked and all typos were corrected, as well as to all scientific names.
- Please remove this section as it has nothing to do with mosquitoes - 5.2.9. SARS-COV-2.
Response: As recommended by the reviewer, the section 5.2.9. SARS-COV-2 was removed from the manuscript.